

# Trends and topics in eye disease research in PubMed from 2010 to 2014

Christophe Boudry[1,2,3], Eric Denion[4], Bruno Mortemousque[5,6] and Fréderic Mouriaux[5,6]

[1] Média normandie, Normandie Université, Université de Caen Normandie, Caen, France
[2] URFIST/Ecole Nationale des Chartes, Paris, France
[3] Laboratoire "Dispositifs d'Information et de Communication à l'Ère Numérique," Conservatoire National des Arts et Métiers, Paris, France
[4] Service d'Ophtalmologie, CHU Côte de Nacre, Caen, France
[5] Service d'Ophtalmologie, CHU Rennes, Université Rennes 1, Rennes, France
[6] Faculté de Médecine, Rennes, France

Corresponding author
Christophe Boudry,
boudry@enc.sorbonne.fr

## ABSTRACT

**Background**: The purpose of this study is to provide a report on scientific production during the period 2010–2014 in order to identify the major topics as well as the predominant actors (journals, countries, continents) involved in the field of eye disease. **Methods**: A PubMed search was carried out to extract articles related to eye diseases during the period 2010–2014. Data were downloaded and processed through developed PHP scripts for further analysis. **Results**: A total of 62,123 articles were retrieved. A total of 3,368 different journals were found, and 19 journals were identified as "core journals" according to Braford's law. English was by far the predominant language. A total of 853,182 MeSH terms were found, representing an average of 13.73 (SD = 4.98) MeSH terms per article. Among these 853,182 MeSH terms, 14,689 different MeSH terms were identified. Vision Disorders, Glaucoma, Diabetic Retinopathy, Macular Degeneration, and Cataract were the most frequent five MeSH terms related to eye diseases. The analysis of the total number of publications showed that Europe and Asia were the most productive continents, and the USA and China the most productive countries. Interestingly, using the mean Five-Year Impact Factor, the two most productive continents were North America and Oceania. After adjustment for population, the overall ranking positions changed in favor of smaller countries (i.e. Iceland, Switzerland, Denmark, and New Zealand), while after adjustment for Gross Domestic Product (GDP), the overall ranking positions changed in favor of some developing countries (Malawi, Guatemala, Singapore). **Conclusions**: Due to the large number of articles included and the numerous parameters analyzed, this study provides a wide view of scientific productivity related to eye diseases during the period 2010–2014 and allows us to better understand this field.

## INTRODUCTION

Bibliometrics has been defined as the use of statistical methods to analyze a body of literature to reveal historical development through the scientific and quantitative study of publications (*Young & Belanger, 1983*). Applications of bibliometrics are numerous: determining a library purchasing policy (*Garfield, 1972*), studying the structure of the network of a scientific field (*Wallace, Larivière & Gingras, 2012*), and mapping the literature related to a research field such as health literacy (*Kondilis et al., 2008*) or cancer research (*Glynn et al., 2010*). Bibliometrics can also evaluate the speed of publication of manuscripts in journals (*Chen, Chen & Jhanji, 2013*), allow one to recognize new topics in the literature (*Boudry, 2015*), or even evaluate research and researchers (*Hirsch, 2005*). Viewed as an objective and quantifiable assessment of research output, such bibliometrics are inextricably linked with research funding, and an investigator must possess a strong publishing background to obtain the financial grants necessary for further work (*Geisler, 2000*).

Few bibliometric studies have appeared in the literature in the field of ophthalmology. Although some analyses have been done on specific countries, (*Kumaragurupari, Sieving & Lalitha, 2010*; *Katibeh, Moein & Javadi, 2011*; *Schlenker, Manalo & Wong, 2013*) continents or territories (*Ugolini et al., 2001*; *Sweileh et al., 2015*), sub-specialties (*Zhao et al., 2011*; *Gupta, Bala & Gupta, 2014*; *Boudry & Mouriaux, 2015*), or production of a selection of ophthalmologic journals (*Kumbar & Akhtary, 1998*; *Guerin et al., 2009*; *Kumar, Cheeseman & Durnian, 2011*; *Liesegang, 2011*), no attempts have been made to describe the field "eye diseases." The main purpose of this paper is to provide a report on scientific production in the field of eye diseases. For this, core journals have been identified, and frequency and distribution of MeSH terms in articles calculated. Furthermore, the geographical distribution and the temporal trend of papers published between 2010 and 2014 have been investigated with ad hoc geographical analyses evaluating literature production and weighted according to major socioeconomic variables, i.e. population size and Gross Domestic Product (GDP).

## MATERIAL AND METHODS

Data were downloaded from PubMed in Extensible Markup Language (XML) and were processed through developed Hypertext Preprocessor language (PHP) scripts, then were imported to Microsoft Excel 2010 (Microsoft, Redmond, USA) for data processing.

### Bibliographic search

The search for papers to be included in this study was carried out on January 16th 2015, using the PubMed database (http://www.ncbi.nlm.nih.gov/pubmed), developed by the National Center for Biotechnology Information (NCBI) at the National Library of Medicine (NLM). PubMed was chosen because it is the most widely used in medicine (*Falagas et al., 2008*). The search strategy was the following: eye diseases [MH] AND journal article [PT] AND 2010:2014 [DP] where MH stands for "Medical Subject Headings," DP "Date of Publication," and PT "Publication Type." "Journal Article" includes the following publication types: journal articles, introductory journal articles, and reviews. MeSH is the National Library of Medicine's controlled vocabulary thesaurus

(http://www.nlm.nih.gov/pubs/factsheets/mesh.html) and consists of sets of terms named "MeSH terms" arranged in a hierarchical structure (called a MeSH tree) with more specific terms arranged beneath less specific terms. In the indexing process, 2 to 50 MeSH terms are assigned to PubMed documents by a team of trained experts (indexers) to properly identify the content, and indexers always index as specifically as possible (*MEDLINE Indexing Online Training Course, 2015*). It is worth mentioning that it is not obligatory for author's keywords to be taken into account by indexers to determine MeSH terms of an article (*Boudry, 2015*, personal data). The MeSH term "Eye Diseases," the definition of which in the NLM-controlled vocabulary thesaurus is "Diseases affecting the eye," was chosen because it covers all diseases of the eye when used in a PubMed search. Indeed, when using a MeSH term in a PubMed search, articles that carry the specified MeSH term are retrieved, but so are all of the articles that carry any of the more specific MeSH terms located beneath it in the tree structure. Thus, searching with the MeSH term "Eye Diseases" automatically includes all the MeSH terms located beneath "Eye Diseases" corresponding to diseases of the eye in general.

## Analysis of MeSH terms

As done in other studies (*Ugolini et al., 2007*, *2010*; *Ramos, González-Alcaide & Bolaños-Pizarro, 2013*; *Neri et al., 2015*), an analysis of the MeSH terms used by PubMed indexers to classify the articles was done by calculating their frequency in the retrieved articles. Check tags, i.e. MeSH terms obligatorily used by indexers to describe recurrent patterns in medical articles, were excluded from this frequency analysis because of their compulsory and recurrent character. They are shown in Supplemental Table S1. The co-occurrence is the simultaneous association of two identical MeSH terms in different articles. The frequency and the percentage of co-occurrence of the MeSH terms located beneath the MeSH term "Eye Diseases" in articles has been calculated. Likewise, the frequency of all other MeSH terms not situated beneath the MeSH term "Eye Diseases" retrieved has also been calculated.

## Analysis of growth of literature

The average yearly growth rate of the literature related to eye diseases was calculated as the mean percentage of annual growth rate for the period studied using the equation: Annual Growth Rate = Current Year Total Number of Articles – Previous Year Total Number of Articles/Previous Year Total Number of Articles (*Deshazo, Lavallie & Wolf, 2009*). The average yearly growth rate was also calculated for the whole PubMed database for the period 2010–2014.

## Bradford's law

Bradford's law has been used extensively in the information science literature to describe the dispersion of articles in any scientific field (*Goffman & Warren, 1969*), or to identify "core journals" of serial titles (*Deshazo, Lavallie & Wolf, 2009*; *Venable et al., 2014*). Bradford's law states that "If scientific journals are arranged in order of decreasing productivity of articles on a given subject, they may be divided into a nucleus of

periodicals more particularly devoted to the subject and several groups or zones containing the same number of articles as the nucleus." To identify the core journals containing articles dedicated to eye diseases, Bradford's law was applied by dividing the journals ranked according to publication output frequency into three groups with each group representing approximately the same number of articles.

## Analysis of journals and language of publication

The 2013 Journal Citation Reports (JCR) (Thomson Reuters, New York, USA) was used for Impact Factor (IF) determination and for the ranking of the journal in the category "Ophthalmology" of the JCR. Language of publication was determined using the "Language" field for all articles retrieved. The percentage of papers published in English in the entire PubMed database between 2010 and 2014 was also calculated using the following search strategy: Medline [sb] OR publisher [sb] OR pubstatusaheadofprint AND 2010:2014 [DP] AND journal article [PT] where sb means "subset."

## Quantitative and qualitative analysis of publication ouput

A quantitative evaluation of the scientific production related to eye diseases was done by evaluating the total number of articles published for each country. To implement a qualitative evaluation, each article was related to the corresponding journal manually in the JCR 2013 to determine the Five-Year Impact Factor (5-Y IF). If a journal did not have an 5-Y IF, the value zero was assigned to the article. For each country, the cumulative 5-Y IF was calculated as the sum of the 5-Y IF of all articles. The mean Five-Year Impact Factor (m5-Y IF) was calculated as the cumulative 5-Y IF divided by the number of articles.

For the quantitative and the qualitative evaluation, we calculated the number of publications and the cumulative and m5-Y IF according to countries and continents.

The country of affiliation of the first author was determined by the address in the "Affiliation" field. When the country was absent in the address, it was determined from the city or email address using an Internet search engine if necessary. If the name of a country no longer existed (e.g. Yugoslavia), the city was used to find the name of the existing country in 2015. England, Scotland, Northern Ireland, and Wales were grouped into the United Kingdom. Hong Kong was considered as part of China.

The GDP is the market value of all officially recognized final goods and services produced within a country in a given period. GDP per capita is often considered an indicator of a country's standard of living (World Bank Group: http://www.worldbank. org/). For each country and each continent, the 5-Y IF GDP index (cumulative 5-Y IF per 1 billion US dollars of GDP) and the 5-Y IF population index (cumulative 5-Y IF per million inhabitants) were calculated using the World Development Indicators from the online databases of the World Bank (http://data.worldbank.org/). The 5-Y IF GDP and the 5-Y IF population index were calculated using the mean GDP and mean number of inhabitants from 2010 to 2013 (data for 2014 were not available). Countries were clustered by their continent according to the United Nations classification (United Nations Statistics Division- Standard Country and Area Codes Classifications: http://unstats.un. org/unsd/methods/m49/m49regin.htm).

## RESULTS

### Journals

The search in PubMed resulted in a total of 62,123 articles over the period 2010 to 2014. We calculated that the average growth rate of literature related to eye diseases was 4.27%. We also calculated that the average growth rate for the whole PubMed database was 6.59% for the same period. Thus, the difference in growth rates between the growth rate of eye diseases and all scientific production in the PubMed database is equal to −2.32 (4.27 − 6.59), which indicates a moderate interest of the scientific community for eye diseases for the studied period, and a stabilization of publication rate (*Michon & Tummers, 2009*).

The 62,123 articles found in the present study were published in 3,368 different journals over the period 2010 to 2014.

Fifty-four journals of 58 belonging to the category "Ophthalmology" in the JCR were found in the 3,368 journals retrieved. These 54 journals published 29,183 articles of the 62,123 (46.98%) that we found in the present study.

According to Bradford's law, the total number of articles compared to the number of journals in order of decreasing productivity was divided into 3 zones containing approximatively the same number of articles. The first third of the total number of articles ($n = 20,779$) represented the journals ($n = 19$; 0.56%) that published between 3,619 and 589 articles (Table 1). The middle third ($n = 20,637$) corresponds to the journals ($n = 118$; 3.5%) that contained an average number of articles (between 290 and 579 articles), and the last third ($n = 20,707$) includes the "long tail" of journals ($n = 3,231$; 95.9%) that published the fewest articles (less than 114). It is important to note that 1,428 journals (42.4%) published only 1 or 2 articles related to eye diseases over the 5-year study period.

### Languages of publication

Thirty-one different languages were identified in the 62,123 articles retrieved. The five predominant languages were English ($n = 55,829$; 89.87%), German ($n = 1,401$; 2.26%), French ($n = 1,132$; 1.82%), Chinese ($n = 927$; 1.49%), and Japanese ($n = 641$; 1.03%). All other languages amounted to less than 1%. The percentage of articles in English in the entire PubMed database was 93.49% for the same period.

### MeSH terms for eye diseases

The MeSH term "Eye Diseases" is located beneath the MeSH term "Diseases" in the MeSH tree. Twenty-seven more specific MeSH terms situated just one level beneath "Eye Diseases" were found in the MeSH tree (Fig. 1A). The MeSH tree beneath "Eye Diseases" includes 6 levels. In total, 243 MeSH terms located beneath "Eye Diseases" were found in these 6 levels. As an example, the MeSH tree for the MeSH term "Chorioretinitis" located six levels beneath "Eye Diseases" is described in Fig. 1B.

Analyzing the 62,123 articles, we found 853,182 MeSH terms, which represent an average of 13.73 (SD = 4.98) MeSH terms per article. Among these 853,182 MeSH terms, 14,689 different MeSH terms were identified: 243 were located beneath "Eye Diseases" and 14,446 were not located beneath "Eye Diseases." The 20 most frequent MeSH terms located in all the levels beneath the MeSH term "Eye Diseases," and for each MeSH term,

**Table 1** Core journals according to Bradford's law (19 most productive journals between 2010 and 2014)

| Journal | Impact Factor[*] (JCR rank [†]) | Language of publication | Number of articles related to "eye diseases" | Total number of articles published by the journal (percentage of articles related to "eye diseases" relative to the total number of articles published) |
|---|---|---|---|---|
| Investigative Ophthalmology & Visual Science | 3.661 (6) | English | 3,619 | 4,955 (73.03) |
| Ophthalmology | 6.170 (2) | English | 1,575 | 1,715 (91.86) |
| British Journal of Ophthalmology | 2.809 (10) | English | 1,319 | 1,496 (88.20) |
| Retina | 3.177 (8) | English | 1,292 | 1,628 (79.36) |
| American Journal of Ophthalmology | 4.021 (5) | English | 1,235 | 1,382 (89.35) |
| Cornea | 2.360 (19) | English | 1,156 | 1,384 (83.52) |
| Journal of Cataract and Refractive Surgery | 2.552 (15) | English | 1,082 | 1,441 (75.10) |
| JAMA Ophthalmology [‡] | 4.488 (3) | English | 1,030 | 1,135 (90.74) |
| PLoS One | 3.534 [§] | English | 1,012 | 82,543 (1.23) |
| Graefe's Archive for Clinical and Experimental Ophthalmology | 2.333 (20) | English | 967 | 1,174 (82.36) |
| Molecular Vision | 2.245 (21) | English | 940 | 1,372 (68.51) |
| Eye | 1.897 (26) | English | 842 | 958 (87.90) |
| Acta Ophthalmologica | 2.512 (16) | English | 780 | 943 (82.71) |
| Optometry and Vision Science | 2.038 (22) | English | 709 | 913 (77.66) |
| Klinische Monatsblätter für Augenheilkunde | 0.665 (53) | German, English | 691 | 858 (80.54) |
| European Journal of Ophthalmology | 1.058 (46) | English | 667 | 738 (90.43) |
| Ophthalmic Plastic and Reconstructive Surgery | 0.914 (50) | English | 647 | 774 (83.59) |
| Journal Français d'Ophtalmologie | 0.361 (56) | French, English | 627 | 688 (91.10) |
| Experimental Eye Research | 3.017 (9) | English | 589 | 1,013 (58.14) |

**Notes:**
JCR, Journal Citation Report.
[*] Impact factor for 2013.
[†] Ranking of journals in the JCR with impact factor in the category "Ophthalmology" (2013). This category includes 58 journals.
[‡] Formerly Archives of Ophthalmology.
[§] JCR Category: Multidisciplinary Sciences.
Journal of Cataract and Refractive Surgery and Ophthalmic Plastic and Reconstructive Surgery also belong in the category "Surgery" of the JCR. Molecular Vision also belongs in the category "Biochemistry and molecular biology" of the JCR.

the 5 most co-occurring MeSH terms in articles are presented in Table 2 (for information on the 21st to 50th see Supplemental Table S2).

Similarly, Table 3 describes the 20 most frequent MeSH terms retrieved in articles and not located beneath "Eye Diseases" (for information on the 21st to 50th see Supplemental Table S3). Among these MeSH terms not located beneath "Eye Diseases," note that some are related to methods of investigation (Tomography, Optical Coherence, Magnetic, Resonance Imaging; Fluorescein Angiography) or type of studies (Retrospective Studies, Follow-Up Studies, Prospective Studies, Cross-Sectional Studies, Case-Control Studies).

## Geographical distribution and socioeconomic factors

The determination of the country of the first author was possible for 59,060 articles (95.07%). One hundred thirty-two different countries were identified. As shown in Table 4A, the qualitative evaluation with the m5-Y IF by continents led to a new ranking compared with the absolute production (total number of publications): North America

**a.**

**All MeSH Categories**
- **Diseases Category**
  - **Eye Diseases**
    - Asthenopia (L1)
    - Cogan Syndrome (L1)
    - Conjunctival Diseases (L1)
    - Corneal Diseases (L1)
    - Eye Abnormalities (L1)
    - Eye Diseases, Hereditary (L1)
    - Eye Hemorrhage (L1)
    - Eye Infections (L1)
    - Eye Injuries (L1)
    - Eye Manifestations (L1)
    - Eye Neoplasms (L1)
    - Eyelid Diseases (L1)
    - Lacrimal Apparatus Diseases (L1)
    - Lens Diseases (L1)
    - Ocular Hypertension (L1)
    - Ocular Hypotension (L1)
    - Ocular Motility Disorders (L1)
    - Optic Nerve Diseases (L1)
    - Orbital Diseases (L1)
    - Pupil Disorders (L1)
    - Refractive Errors (L1)
    - Retinal Diseases (L1)
    - Scleral Diseases (L1)
    - Uveal Diseases (L1)
    - Vision Disorders (L1)
    - Vitreoretinopathy, Proliferative (L1)
    - Vitreous Detachment (L1)

**b.**

**Eye Diseases**
- Uveal Diseases (L1)
  - Uveitis (L2)
    - Panuveitis (L3)
      - Uveitis, Posterior (L4)
        - Choroiditis (L5)
          - Chorioretinitis (L6)

**Figure 1** (A) MeSH term "Eye Diseases" and the 27 MeSH terms located just one level beneath. (B) Example of the MeSH term "Chorioretinitis." L1 to L6: Level 1 to Level 6 in the MeSH tree beneath the MeSH term "Eye Diseases."

and Oceania initially 3$^{rd}$ and 4$^{th}$, were respectively first and second when considering m5-Y IF. Geographic distribution by continent of publication was also different after adjustment for 5-Y IF population index and 5-Y IF GDP index (Tables 4A and 4C).

When analyzing countries, the United States was the most absolute productive country for total number of publications (Table 5A). Among the most productive countries, 15 of 20 were classified as developed countries (according to the definition of the United Nations). Qualitative evaluation using m5-Y IF highlighted the quality of the scientific production related to eye diseases of some countries (i.e. United kingdom, the

**Table 2 Twenty most frequent MeSH terms retrieved in articles and located in all levels beneath the MeSH term "Eye Diseases" in the MeSH tree and co-occurrence of MeSH terms in articles**

| MeSH terms located beneath "Eye Diseases" | Number of articles indexed with this MeSH term (%) | Five most co-occurring MeSH terms in articles with the MeSH term in the first column (% of co-occurrence) |
|---|---|---|
| Vision Disorders[L1] | 3,471 (5.59) | Visual Acuity (34.8)<br>Visual Fields (19.25)<br>Retrospective studies (12.79)<br>Treatment Outcome (12.22)<br>Tomography, Optical Coherence (11.93) |
| Glaucoma[L2] | 3,145 (5.06) | Intraocular Pressure (48.51)<br>Retinal Ganglion Cells (17.24)<br>Visual Fields (15.78)<br>Tomography, Optical Coherence (15.14)<br>Tonometry, Ocular (15.01) |
| Diabetic Retinopathy[L2] | 3,084 (4.96) | Diabetes Mellitus, Type 2 (22.86)<br>Macular Edema (21.77)<br>Visual Acuity (18.32)<br>Retina (17.67)<br>Tomography, Optical Coherence (13.59) |
| Macular Degeneration[L3] | 2,756 (4.44) | Visual Acuity (35.81)<br>Antibodies Monoclonal Humanized (27.58)<br>Angiogenesis Inhibitors (26.96)<br>Tomography, Optical Coherence (25.91)<br>Vascular Endothelial Growth Factor A (21.52) |
| Cataract[L2] | 2,656 (4.28) | Visual Acuity (30.27)<br>Cataract Extraction (26.88)<br>Lens Implantation Intraocular (21.99)<br>Lens Crystalline (18.19)<br>Phacoemulsification (17.09) |
| Retinal Diseases[L1] | 2,138 (3.44) | Tomography, Optical Coherence (24.84)<br>Retina (22.26)<br>Visual Acuity (20.39)<br>Fluorescein Angiography (16.00)<br>Retrospective Studies (12.35) |
| Myopia[L2] | 2,136 (3.44) | Visual Acuity (38.13)<br>Refraction Ocular (31.76)<br>Keratomileusis Laser In Situ (23.47)<br>Lasers Excimer (19.48%)<br>Retrospective Studies (18.31) |
| Glaucoma, Open-Angle[L3] | 2,006 (3.23) | Intraocular Pressure (61.22)<br>Tonometry, Ocular (22.98)<br>Visual Fields (22.33)<br>Prospective Studies (21.09)<br>Antihypertensive Agents (17.05) |
| Eye Diseases | 2,005 (3.23) | Visual Acuity (11.47)<br>Retrospective Studies (11.27)<br>Eye Diseases, Hereditary (10.12)<br>Ophthalmology (9.08)<br>Eye (8.33) |

| Table 2 Continued | | |
|---|---|---|
| **MeSH terms located beneath "Eye Diseases"** | **Number of articles indexed with this MeSH term (%)** | **Five most co-occurring MeSH terms in articles with the MeSH term in the first column (% of co-occurrence)** |
| Blindness[(L2)] | 1,810 (2.91) | Visual Acuity (16.46)<br>Prevalence (11.93)<br>Magnetic Resonance Imaging (10.22)<br>Visually Impaired Persons (9.72)<br>Risk Factors (8.12) |
| Corneal Diseases[(L1)] | 1,599 (2.57) | Visual Acuity (28.33)<br>Cornea (23.08)<br>Retrospective Studies (21.08)<br>Treatment Outcome (19.20)<br>Epithelium Corneal (14.88) |
| Sjogren's Syndrome[(L3)] | 1,410 (2.27) | Salivary Glands (14.89)<br>Autoantibodies (13.05)<br>Lupus Erythematosus, Systemic (11.21)<br>Treatment Outcome (9.36)<br>Case-Control Studies (8.94) |
| Macular Edema[(L4)] | 1,399 (2.25) | Visual Acuity (52.75)<br>Tomography, Optical Coherence (49.46)<br>Diabetic Retinopathy (46.89)<br>Intravitreal Injections (28.73)<br>Treatment Outcome (28.31) |
| Retinal Detachment[(L2)] | 1,396 (2.25) | Visual Acuity (42.91)<br>Vitrectomy (36.32)<br>Retrospective Studies (29.87)<br>Tomography, Optical Coherence (26.2)<br>Treatment Outcome (21.56) |
| Optic Nerve Diseases[(L1)] | 1,237 (1.99) | Optic Disk (39.45)<br>Intraocular Pressure (33.23)<br>Retinal Ganglion Cells (31.85)<br>Tomography, Optical Coherence (30.23)<br>Visual Fields (29.59) |
| Uveitis[(L2)] | 1,181 (1.90) | Treatment Outcome (18.37)<br>Retrospective Studies (18.29)<br>Visual Acuity (16.93)<br>Autoimmune Diseases (11.85)<br>Disease Models, Animal (10.84) |
| Choroidal Neovascularization[(L3)] | 1,137 (1.83) | Angiogenesis Inhibitors (41.60)<br>Visual Acuity (40.90)<br>Antibodies, Monoclonal, Humanized (39.93)<br>Fluorescein Angiography (38.70)<br>Macular Degeneration (36.50) |
| Behcet Syndrome[(L5)] | 1,106 (1.78) | Treatment Outcome (17.27)<br>Case-Control Studies (14.29)<br>Immunosuppressive Agents (13.29)<br>Retrospective Studies (11.66)<br>Genetic Predisposition to Disease (9.95) |
| Retinal Degeneration[(L2)] | 1,063 (1.71) | Retina (33.30)<br>Disease Models Animal (31.89)<br>Electroretinography (20.70)<br>Photoreceptor Cells Vertebrate (18.44)<br>Mice Inbred C57BL (17.87) |

(Continued)

| MeSH terms located beneath "Eye Diseases" | Number of articles indexed with this MeSH term (%) | Five most co-occurring MeSH terms in articles with the MeSH term in the first column (% of co-occurrence) |
|---|---|---|
| Dry Eye Syndromes[(L2)] | 1,051 (1.69) | Tears (46.91) Ophthalmic Solutions (20.74) Questionnaires (17.79) Prospective Studies (17.32) Cornea (15.98) |

**Note:**
[(L1), (L2), (L3), (L4), (L5)]: Levels in the MeSH tree beneath the MeSH term "Eye Diseases," respectively first, second, third, fourth, and fifth levels.

**Table 3 Twenty most frequent MeSH terms retrieved in articles and not located beneath the MeSH term "Eye Diseases" in the MeSH tree**

| MeSH terms | Number of articles indexed with this MeSH term (%) |
|---|---|
| Visual Acuity | 10,938 (17.60) |
| Retrospective Studies | 8,333 (13.41) |
| Treatment Outcome | 8,078 (13.00) |
| Follow-Up Studies | 5,913 (9.52) |
| Tomography, Optical Coherence | 5,817 (9.36) |
| Prospective Studies | 5,264 (8.47) |
| Intraocular Pressure | 4,185 (6.74) |
| Risk Factors | 3,981 (6.41) |
| Retina | 3,771 (6.07) |
| Disease Models Animal | 3,141 (5.06) |
| Magnetic, Resonance Imaging | 3,072 (4.94) |
| Diagnosis Differential | 2,935 (4.72) |
| Fluorescein Angiography | 2,923 (4.70) |
| Cornea | 2,745 (4.42) |
| Postoperative Complications | 2,654 (4.27) |
| Cross-Sectional Studies | 2,390 (3.85) |
| Time Factors | 2,295 (3.69) |
| Case-Control Studies | 2,278 (3.67) |
| Visual Fields | 2,201 (3.54) |
| Prevalence | 2,177 (3.50) |

Netherlands, and Singapore). After adjustment for population size (Table 5B), the United States was downgraded to 9th place, and China disappeared from the top 20 countries. The ranking positions changed overall in favor of smaller countries with a low population size, with the appearance of Iceland, Switzerland, Denmark, and New Zealand in the top 10. When normalized by GDP (Table 5C), 10 of the top 20 countries in terms of absolute production (number of articles produced), disappeared (i.e. China, Japan, Germany). Furthermore, the ranking positions changed in favor of some developing countries (i.e. Malawi, Guatemala, Singapore ). Nine countries appeared in the top 25 countries for the

**Table 4 Production of articles related to eye diseases, by continent (2010–2014).** (A) Based on the total number of publications, (B) Based on the 5-Y IF population index (cumulative 5-Y IF per million inhabitants), (C) Based on the 5-Y IF GDP index (cumulative 5-Y IF per 1 billion US dollars of GDP)

| A | | | | B | | C | |
|---|---|---|---|---|---|---|---|
| Continent | No. of articles (%) | Cumulative 5-Y IF (sum of 5-Y IF) | m5-Y IF (mean 5-Y IF) | Continent | 5-Y IF population index | Continent | 5-Y IF GDP index |
| Europe | 19,716 (33.38) | 52118.511 | 2.64 | North America | 166.2 | Oceania | 3.69 |
| Asia | 18,140 (30.71) | 39656.153 | 2.19 | Oceania | 158.41 | North America | 3.28 |
| North America | 16,643 (28.18) | 57741.761 | 3.47 | Europe | 70.57 | Europe | 2.49 |
| Oceania | 1,919 (3.25) | 5877.951 | 3.06 | Asia | 9.47 | Asia | 1.74 |
| Latin America and the Caribbean | 1,560 (2.64) | 2740.639 | 1.76 | Latin America and the Caribbean | 4.52 | Africa | 0.73 |
| Africa | 1,082 (1.83) | 1578.21 | 1.46 | Africa | 1.47 | Latin America and the Caribbean | 0.47 |
| World | 59,060 (100) | 159713.225 | 2.70 | World | 22.8 | World | 2.23 |

Note:
GDP, gross domestic product; 5-Y IF, five-year impact factor; m5-Y IF, mean five-year impact factor.

**Table 5 Top twenty countries for publications related to eye diseases (2010–2014).** (A) Based on the total number of publications, (B) Based on the 5-Y IF population index (cumulative 5-Y IF per million inhabitants), (C) Based on the 5-Y IF GDP index (cumulative 5-Y IF per 1 billion US dollars of GDP)

| A | | | | B | | C | |
|---|---|---|---|---|---|---|---|
| Country | No. of articles (%) | Cumulative 5-Y IF (sum of 5-Y IF) | m5-Y IF (mean 5-Y IF) | Country | 5-Y IF population index | Country | 5-Y IF GDP Index |
| United States | 15,266 (25.85) | 53328.643 | 3.49 | Iceland | 567.11 | Iceland | 12.65 |
| China | 4,967 (8.41) | 11319.11 | 2.28 | Singapore | 376.18 | Malawi | 11.43 |
| United Kingdom | 4,141 (7.01) | 14502.337 | 3.50 | Switzerland | 246.27 | Guatemala | 10.90 |
| Japan | 4,029 (6.82) | 9644.649 | 2.39 | United Kingdom | 228.55 | Singapore | 7.20 |
| Germany | 3,247 (5.5) | 7569.76 | 2.33 | Australia | 228.41 | Israel | 6.16 |
| India | 2,288 (3.87) | 4259.227 | 1.86 | Denmark | 205.73 | United Kingdom | 5.64 |
| Italy | 2,200 (3.73) | 6055.104 | 2.75 | Israel | 204.07 | Greece | 4.46 |
| Korea. Rep. | 2,137 (3.62) | 5082.929 | 2.38 | Netherlands | 192.01 | New Zealand | 4.35 |
| France | 2,045 (3.46) | 4896.101 | 2.39 | United States | 170.53 | Korea. Rep. | 4.21 |
| Spain | 1,820 (3.08) | 4115.654 | 2.26 | New Zealand | 163.37 | Tunisia | 4.16 |
| Turkey | 1,682 (2.85) | 2573.658 | 1.53 | Austria | 140.55 | Nepal | 3.96 |
| Australia | 1,669 (2.83) | 5152.254 | 3.09 | Finland | 130.29 | Netherlands | 3.77 |
| Canada | 1,377 (2.33) | 4413.118 | 3.20 | Sweden | 128.62 | Australia | 3.66 |
| Brazil | 1,005 (1.7) | 1690.9 | 1.68 | Canada | 127.68 | Denmark | 3.48 |
| Netherlands | 861 (1.46) | 3209.807 | 3.73 | Greece | 108.60 | United States | 3.36 |
| Switzerland | 757 (1.28) | 1958.809 | 2.59 | Korea. Rep. | 101.96 | Turkey | 3.30 |
| Iran. Islamic Rep. | 644 (1.09) | 1213.767 | 1.88 | Italy | 101.75 | Switzerland | 2.98 |
| Poland | 627 (1.06) | 658.179 | 1.05 | Norway | 98.12 | Spain | 2.90 |
| Israel | 589 (1) | 1599.869 | 2.72 | Germany | 93.27 | Hungary | 2.89 |
| Singapore | 550 (0.93) | 1972.32 | 3.59 | Spain | 88.16 | Serbia | 2.89 |

Note:
GDP, gross domestic product; 5-Y IF, five-year impact factor; m5-Y IF, mean five-year impact factor.
three parameters studied: the United States, United Kingdom, the Republic of Korea, Spain, Australia, the Netherlands, Switzerland, Israel and Singapore.

## DISCUSSION

To our knowledge, there are no similar studies examining worldwide research related to ophthalmology, and such an analysis of articles related to eye diseases has never been done. The MeSH term "Eye diseases" was chosen to retrieve articles in PubMed because it reflects an extensive coverage of eye diseases. Although it would have been easier and less complex to analyze MeSH terms without considering their position in the MeSH tree (located or not beneath eye diseases), we showed that separate analysis was more informative than not considering the position. In the former case, we would have identified only 5 eye diseases (Vision Disorders, Glaucoma, Diabetic Retinopathy, Macular Degeneration, and Cataract). In contrast, including the position in the MeSH tree allowed us to identify the 20 most frequent topics related to eye diseases such as diagnostic techniques, physiological phenomena, anatomic structures, and methods of investigation. We also performed an analysis of co-occurring terms for the 20 most frequent MeSH terms located beneath the MeSH term "Eye Diseases." To the best of our knowledge, such an analysis has never been performed, and no study has defined, individually, the topics related to the 20 most frequent eye diseases in the literature over the 5-year study period. We could have chosen another MeSH term upon which to build our query in PubMed for this analysis: "Ophthalmology," whose definition in the NLM-controlled vocabulary thesaurus is "A surgical specialty concerned with the structure and function of the eye and the medical and surgical treatment of its defects and diseases." Using this MeSH term instead of "Eye Diseases" over the 5-year study period would have resulted in the analysis of only 734 articles, instead of the 62,123 articles included in the present study. This shows that the MeSH term "Ophthalmology" is not adequate for retrieving articles in the field of ophthalmology using PubMed.

Our study proposes a quantitative evaluation, i.e. the number of articles published by a journal and a qualitative evaluation based on the 5-Y IF, which takes into account the average number of citations of articles published by a journal in a 5-year period. Note that using the 5-Y IF is more qualitative, but does not reflect individual citations of articles. PubMed was chosen because of its open access, broad coverage, international visibility, quality criteria, and because it uses a controlled vocabulary thesaurus for indexing and retrieving documents (*Ramos, González-Alcaide & Bolaños-Pizarro, 2013*). However our methodology has several limitations: 1) PubMed is the most widely used database for bibliometric analysis, but it does not contain all biomedical journals and is biased in favor of English journals (*Ugolini et al., 2007*; *Vioque et al., 2009*), 2) The methodology for identifying authors' country affiliations did not allow us to assess the country for all articles studied. Furthermore, because the PubMed database indexes only the affiliation of the first author before the year 2014, when present the country affiliation of the authors indicated only one country per article and fails to identify collaborative research, 3) Attributing the credit of articles completely to the first author may also not always indicate the country in which the research was conducted, and consequently may lead to underestimation of the

research output in developed countries. Indeed, the first author, in some cases, may be a student from a developing country doing short-term research training in a developed country, whereas the last author is most often the person who supervises and provides the most funding support for the project in a developed country. Despite these limitations and because 62,123 articles were included, we believe that this study provides a wide view of scientific productivity related to the field of ophthalmology during the period 2010–2014. Moreover, since a part of the methodology used for this study has been used by other authors, some of our results may be compared with those of others in the future (*Ramos et al., 2008*; *Vioque et al., 2009*; *Ramos, González-Alcaide & Bolaños-Pizarro, 2013*).

The articles retrieved in this study were published in 3,368 journals, including 54 of 58 journals indexed in the category "Ophthalmology" in the JCR. Logically, 18 of 19 of the core journals according to Bradford's law were included in the category "Ophthalmology" in the JCR. PLoS One, belonging to the JCR category "Multidisciplinary sciences," has published the lowest percentage of articles related to eye diseases relative to the total number of articles produced (1.23%). For the other journals, the percentage of articles related to eye diseases relative to the total number of articles produced varied from 58.14% (Experimental Eye research) to 91.86% (Ophthalmology). We found that no journal attained 100% articles related to eye diseases because journals also publish articles related to investigational methods, basic science such as cell biology, molecular biology, and cellular experimental topics. Although these results may seem logical, we did not find similar results in the literature.

The United States, as expected, regardless of the scientific specialty or sub-specialty considered, was by far the most absolute productive country. However, when comparing our results with former studies in Ophthalmology (*Ohba, 2005*; *Guerin et al., 2009*), China experienced spectacular growth and is now the second most absolute productive country. Unexpectedly, the Islamic Republic of Iran appeared in the top 20 absolute productive countries in ophthalmology which may be explained by the high annual average growth rate of 67% of Iranian publications related to ophthalmology during the period 2001 to 2010 (*Katibeh, Moein & Javadi, 2011*). The qualitative evaluation using m5-Y IF did not reflect the absolute production (number of articles produced), demonstrating the usefulness of considering this parameter for evaluating scientific production of countries. When normalized by population, the overall ranking positions changed in favor of developed countries and also in favor of smaller countries with a small population size. Furthermore, as found in other studies (*Ugolini et al., 2001*; *Robert et al., 2004*; *Ohba, 2005*; *Albayrak et al., 2012*; *Boudry & Mouriaux, 2015*), Nordic European countries were efficient when adjustment was made for population. When normalized by GDP, the ranking positions changed partially in favor of developing countries. The reason for this observation is difficult to interpret, but a better utilization of resources may be the reason (*Ugolini et al., 2007*).

### Funding

The authors received no funding for this work.

## Competing Interests

The authors declare that they have no competing interests.

## Author Contributions

- Christophe Boudry conceived and designed the experiments, performed the experiments, analyzed the data, contributed reagents/materials/analysis tools, wrote the paper, prepared figures and/or tables, reviewed drafts of the paper.
- Eric Denion reviewed drafts of the paper.
- Bruno Mortemousque reviewed drafts of the paper.
- Fréderic Mouriaux conceived and designed the experiments, analyzed the data, contributed reagents/materials/analysis tools, wrote the paper, prepared figures and/or tables, reviewed drafts of the paper.

## Data Deposition

The research in this article did not generate any raw data.

## Supplemental Information

Supplemental information for this article can be found online at http://dx.doi.org/10.7717/peerj.1557#supplemental-information.

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
