# Peer review of "Trends and topics in eye disease research in PubMed from 2010 to 2014"

_PeerJ, doi:10.7717/peerj.1557_

## Round 0.1 · original submission · Major Revisions

You will see that both reviewers asked for major revisions.

Reviewer 1 ·

Basic reporting

Please refer to "Comments for the Author" below.

Experimental design

Please refer to "Comments for the Author" below.

Validity of the findings

Please refer to "Comments for the Author" below.

Additional comments

General comments:
The overall findings of this bibliometric analyses by Boudry et al. are interesting, which has comprehensively included all the eye disease-related papers published between 2010 and 2014. The manuscript is mostly well written, and the tables and figures are fairly legible. The results provide first-hand statistic data revealing the past situation in the research field of ophthalmology and visual science. Nevertheless, I have two major concerns about the criteria used in the analyses, which may lead to misinterpretation of the data. Accordingly, I strongly recommend that the authors delve into the data further, so as to make their conclusion strengthened and more informative to readers.

Major comments:
1. The authors attempted to correlate the population or GPA with the production of literatures related to eye diseases. However, the reliability of their conclusions depends substantially on the accurate identification of the true ownership of these papers. It is well-acknowledged that the person whose name listed as the first position in the byline is mostly likely the one that conducts most experiments, while the person whose name placed at the last position (or the last corresponding author in the byline) is the one that supervises and provides the most funding supports for the project (under most circumstances this applies to the collaborative work, as well). I personally have come across many cases that students from developing countries went to developed countries for a short-term training in research. As these students were financially supported by exchange programs of their native countries, to observe the policy on these exchange activities and meet the degree requirement on publication, their first affiliations in the later publications were unexceptionally written as their native countries, even though the true ownership (and majority of the research cost) should be credited to principle investigators who were residing in developed countries. Under such circumstances, it is inappropriate to attribute the credit completely to the first authors; otherwise, the research output in developed countries would be much underestimated. Considering that many papers nowadays have several co-first or co-corresponding authors, and that each person in the meantime may have multiple affiliations, to reduce the workload, I believe the better way to deal with this issue is that in addition to applying the current criterion that the country affiliation is the first affiliation of the first author, the current study should also adopt another criterion that the country affiliation is the first affiliation of the last author (or last co-corresponding author) in the byline, followed by comparing these two correlation results. If the first author is also a co-corresponding author, the new criterion could prescribe that the country affiliation remains to be the first affiliation of the first author. The re-analysis will not take too much time, but will make the conclusion more accurate and informative.

2. The authors here applied Bradford’s Law to define the core journals which they consider to be “…of the highest interest for researchers interested in eye diseases…”, whereas “the last third should be regarded as of less importance for eye disease researchers during the 5-year study period”. Such a conclusion is problematic. First, it is much harder to get published in high-profile journals (like Nature, Lancet, etc); nevertheless, once published, these papers are always drawing greater attention than those with low impact factors. Therefore, the journal which publishes the fewest eye disease-related papers is not necessarily unimportant. In contrast, some of the core journals defined by Bradford’s Law here are rarely cited (indicated by the very low impact factors); these journals are obviously not of the highest interest of researchers in ophthalmology and visual field. Second, projects that get published in high-profile journals usually cost more money, take longer times, and involve more researchers. Thus, the term of productivity here is misleading, not to mention that such the numbers of publications were later used to calculate the ranking positions adjusted for population and GDP. To get the accurate citation of each of these 62123 papers is extremely time-consuming; hence, using the five-year impact factor of a journal to qualitatively estimate the productivity is preferred, even though a specific paper may not receive exactly the same attention as the journal per se (but it is undoubted that a paper of low quality could barely get accepted by top journals). Comparing the ranking positions of a country based on the numbers of papers and the quality of papers (i.e. using the sum of impact factors) would yield more information (e.g. the quality of research), especially after GDP and population adjustments. Because this analysis won’t take much time, but will enormously improve the quality of the current work, I strongly recommend the authors to further mine their data.

Minor comment:
The manuscript requires the further proofreading. For example, Line 30, “6 2123 articles…” and “3 368 different journals”, please make the format of the digits consistent. Another example is Line 144, should there be a “[sb]” between “pubstatusaheadofprint” and “AND”?

Reviewer 2 ·

Basic reporting

In this manuscript, Boudry and colleagues analysed publications in ocular disease from 2010 to 2015. This study is well conducted with convincing methods and results. However, several concerns were raised based on this study.

1. The "core journals according to Bradford's law" were only limited to ophthalmology. However, many high profile journals like comprehensive journals including Science, Nature, PNAS, and neuroscience journals were not included. It would neglect the most important studies in eye diseases.

2. The topics were restricted in Diseases category -> Eye diseases, which might not be a good way. For example, the most prevalent blindness leading diseases AMD, glaucoma and retinitis pigmentosa are not included in this categories. Obviously, they are eye diseases for sure, just not fall in "Diseases category -> Eye diseases" category. They might in neurodegenerative disease or neuroscience category. So, the ocular diseases not fall in "Diseases category -> Eye diseases" category should be included as important parts of eye diseases.

3. In this manuscript, the authors cited by a (author + year) formation. Usually, it should be properly before the punctuations. However, most of the citations in this manuscript is as "policy,(Garfield & others, 1972)". The formation of citation should be corrected.

Experimental design

Please see "Basic Reporting".

Validity of the findings

Please see "Basic Reporting".

---

## Round 0.2 · accepted · Accept

The reviewers have agreed to your revision.

Reviewer 2 ·

Basic reporting

The authors have revised their manuscript with significant improvement. They answers almost all my questions and made changes adequately. So I suggest PeerJ accept this manuscript as a research article.

Experimental design

Please see "Basic Reporting".

Validity of the findings

Please see "Basic Reporting".

Additional comments

Please see "Basic Reporting".